# Metabolomics: A New Era in the Diagnosis or Prognosis of B-Cell Non-Hodgkin’s Lymphoma

**DOI:** 10.3390/diagnostics13050861

**Published:** 2023-02-23

**Authors:** Abdullah Alfaifi, Mohammed Y. Refai, Mohammed Alsaadi, Salem Bahashwan, Hafiz Malhan, Waiel Al-Kahiry, Enas Dammag, Ageel Ageel, Amjed Mahzary, Raed Albiheyri, Hussein Almehdar, Ishtiaq Qadri

**Affiliations:** 1Department of Biological Science, Faculty of Science, King Abdulaziz University, Jeddah 21589, Saudi Arabia; 2Fayfa General Hospital, Ministry of Health, Jazan 83581, Saudi Arabia; 3Department of Biochemistry, College of Science, University of Jeddah, Jeddah 21493, Saudi Arabia; 4Hematology Research Unit, King Fahad Medical Research Center, King Abdulaziz University, Jeddah 21589, Saudi Arabia; 5Department of Hematology, Faculty of Medicine, King Abdulaziz University, Jeddah 21589, Saudi Arabia; 6King Abdulaziz University Hospital, King Abdulaziz University, Jeddah 21589, Saudi Arabia; 7Prince Mohammed Bin Nasser Hospital, Ministry of Health, Jazan 82943, Saudi Arabia; 8Eradah Hospital, Ministry of Health, Jazan 82943, Saudi Arabia

**Keywords:** metabolomics, B-cell non-Hodgkin’s lymphoma, biomarkers, metabolites, early diagnosis, therapeutic

## Abstract

A wide range of histological as well as clinical properties are exhibited by B-cell non-Hodgkin’s lymphomas. These properties could make the diagnostics process complicated. The diagnosis of lymphomas at an initial stage is essential because early remedial actions taken against destructive subtypes are commonly deliberated as successful and restorative. Therefore, better protective action is needed to improve the condition of those patients who are extensively affected by cancer when diagnosed for the first time. The development of new and efficient methods for early detection of cancer has become crucial nowadays. Biomarkers are urgently needed for diagnosing B-cell non-Hodgkin’s lymphoma and assessing the severity of the disease and its prognosis. New possibilities are now open for diagnosing cancer with the help of metabolomics. The study of all the metabolites synthesised in the human body is called “metabolomics.” A patient’s phenotype is directly linked with metabolomics, which can help in providing some clinically beneficial biomarkers and is applied in the diagnostics of B-cell non-Hodgkin’s lymphoma. In cancer research, it can analyse the cancerous metabolome to identify the metabolic biomarkers. This review provides an understanding of B-cell non-Hodgkin’s lymphoma metabolism and its applications in medical diagnostics. A description of the workflow based on metabolomics is also provided, along with the benefits and drawbacks of various techniques. The use of predictive metabolic biomarkers for the diagnosis and prognosis of B-cell non-Hodgkin’s lymphoma is also explored. Thus, we can say that abnormalities related to metabolic processes can occur in a vast range of B-cell non-Hodgkin’s lymphomas. The metabolic biomarkers could only be discovered and identified as innovative therapeutic objects if we explored and researched them. In the near future, the innovations involving metabolomics could prove fruitful for predicting outcomes and bringing out novel remedial approaches.

## 1. Introduction

B-cell non-Hodgkin’s lymphomas (B-NHLs) are a genetically, metabolically, and clinically heterogeneous group of neoplasms, with most emerging from B lymphocytes in the germinal centre (GC). B-NHLs account for approximately 90% of all non-Hodgkin’s lymphomas [1]. Diffuse large B-cell lymphomas (DLBCLs), follicular lymphoma (FL), Burkitt lymphoma (BL), and B-cell chronic lymphocytic leukaemia/small lymphocytic lymphoma (CLL/SLL) are typical B-NHL subtypes [2]. Myc amplification [3] and metabolic heterogeneity in B-NHL are important biologically because they influence therapy responses and can predict clinical outcomes [4,5].

As cells are driven to grow, proliferate, or die, their metabolic needs fluctuate, and it is essential that cellular metabolism correspond to these needs [6]. B-cell lymphoma and cancer cells have dysregulated metabolisms that promote uncontrolled proliferation [7,8]. This altered metabolism leads to metabolic phenotypes that can be utilised for earlier cancer detection and/or therapy response biomarkers [9]. Fluorodeoxyglucose-PET imaging is an essential tool for the management of many malignancies, including B-cell lymphomas [10]. Other metabolites in biological samples have been in the limelight for diagnosis, monitoring, and therapy [11].

Metabolomics is a comprehensive evaluation of both qualitative and quantitative parameters of all the metabolites present in cells, tissues, and bodily fluids, which can reveal crucial information about the cancer state that would not be obvious otherwise. Metabolomics-based diagnosis investigates the metabolites present in the human body and how they react under stress conditions, like various diseases and disorders [12,13]. Metabolomics is a powerful tool that can identify cancer biomarkers and drivers of tumorigenesis. An example includes the de novo synthesis of phospholipid compounds in malignant tissues, which increases at the time of the progression of the tumour [14,15]. Worthy, LDH-A was the first metabolic target demonstrated to be directly regulated by an oncogene (MYC), and genetic or pharmacologic inhibition of LDH-A diminishes MYC-dependent tumours [16]. Even now, it is a challenging task to detect and treat the lymphoma at an initial stage.

This review provides an overview of existing and future metabolomics prospects to improve B-cell non-Hodgkin’s lymphoma diagnosis, monitoring, and treatment. First, we review B-cell non-Hodgkin’s lymphoma metabolism. We then introduce general metabolomics techniques, including their analytical advantages and disadvantages. In the final section, we present instances where metabolomics has been employed in the clinical and research areas as a way to lead prospective future applications for the prognosis and diagnosis of B-cell non-Hodgkin’s lymphoma. In practice, metabolomics has been widely covered. For more information on best practises in metabolomics analysis, the reader is referred to other excellent reviews cited throughout this article.

## 2. Metabolism in B-Cell Non-Hodgkin’s Lymphoma (B-NHL)

Cell metabolism is a well-defined set of metabolic activities that generate and store energy equivalents, maintain redox homeostasis, synthesise biologically active macromolecules, and eliminate organic waste [17]. Catabolism breaks down carbon sources into simpler intermediates, which are then employed as building blocks in the production of lipids, amino acids, carbohydrates, and nucleotides (anabolism) [18]. Tumour cells are able to survive, grow, and divide because of their metabolic versatility and plasticity, which allow them to produce ATP as an energy source while maintaining the reduction–oxidation (redox) balance and devoting resources to biosynthesis [19]. Recent sequencing approaches have not discovered significant metabolic genes as direct lymphoma driver mutations (Figure 1) [20,21].

Metabolic alterations in B-NHL are characterised by the production of enough energy and maintenance of anabolism for survival, growth, and division in the face of low levels of nutrients and oxygen (such as HIF1 and MYC), deregulation of metabolic regulators (like mTORC1), and rewiring of metabolic pathways (e.g., BCR signalling) [22,23].

The Warburg effect promotes aerobic glycolysis over aerobic oxidation [24], and this is supported by HIF1-alpha and MYC. This leads to the production of lactate and poor producing ATP, but helps create biomass. As a result, the body’s reaction to hypoxia-induced metabolic abnormalities may promote anabolism in GC-derived B-cell lymphoma [22].

MYC oncogene aberrations, including translocations or overexpression, are characteristics of B-cell lymphoma aetiology [25]. B-cell lymphomas require higher MYC levels to maintain their rapid proliferation rate. MYC upregulates nucleoside metabolism, which is essential for cell development. Glutamine metabolism is similarly regulated by MYC expression [25,26]. Glucose uptake, glycolysis, and lipid biosynthesis are all controlled by MYC as well [27]. On the other hand, alpha-ketoglutarate (αKG) synthesis can be inhibited by hypoxia and mitochondrial dysfunction, which in turn reduces the activity of αKG-dependent enzymes, leading to increased DNA and histone hypermethylation and stabilisation of HIF1α. HIF1α is the primary transcriptional regulator of the adaptive response to hypoxia and is constitutively stabilised in a significant proportion of DLBCLs and FLs [22]. HIF1α and MYC promote anaerobic glycolysis by activating genes for glucose transporter (GLUT), hexokinase (HK), monocarboxylate transporter (MTC), pyruvate dehydrogenase (PDK), phosphofructokinase (PFK), phosphoglycerate kinase (PGK), pyruvate kinase (PK), and lactate dehydrogenase (LDHA) [27].

mTORC1 is essential for generating metabolic precursors via the tricarboxylic acid cycle (TCA) and stimulating cellular proliferation. Activation of mTORC1 thereby enhances the survival of B-cell lymphoma. T-cell-selected GC B cells in the light zone necessitate mTORC1 activation in order to proliferate and mutate in the dark zone. mTORC1 may be aberrantly activated in GCB-DLBCL through activating mutations of PI3K/Akt/mTOR pathway genes [22].

A further marker of B-cell lymphoma is altered B-cell receptor (BCR) signalling, which is essential for the maintenance and creation of both healthy and malignant B cells [28]. PI3K/AKT/mTORC1 is one of the BCR signalling pathway’s downstream branches. PI3K regulates glycolysis and energy generation, and consequent AKT signalling influences the cellular metabolome. AKT promotes glucose uptake and glycolysis by increasing the expression and translocation of GLUT1 and glycolytic enzymes, including hexokinase (HK) expression and activation [28].

In a subset of DLBCL and MCL, PTEN mutations lead to AKT/mTORC1 pathway gene expression [29]. RagC mutations in FL enhance mTORC1 signalling by eliminating amino acid dependence [30]. Numerous anabolic and energy-generating processes, including protein synthesis, pyrimidine synthesis, HIF1α expression, glycolysis, the oxidative portion of the pentose phosphate pathway (PPP), lipid and mitochondrial metabolism, and glutaminolysis, are stimulated by mTORC1 expression [23].

There is an urgent need for biomarkers based on non-invasive sampling procedures (e.g., blood, urine, etc.) that can help in the diagnosis of lymphoma, such as metabolite profiling. The perfect test should be easy, reliable, and accurate. “What simple, non-invasive, painless, and convenient tests can be used to detect cancer early?” ranked as the most important research priority for the early detection of cancer in the UK-focused research gap survey performed by the James Lind Alliance, which includes patients and doctors [31]. Accordingly, serum biomarkers of lymphoma activity have been studied extensively over the last decade [32], and we conclude that they are clinically relevant for the diagnosis, prognosis, and therapeutic monitoring of lymphomas. In this review, we shed light on the major metabolic dysregulation described in B-cell non-Hodgkin’s lymphoma research (Table 1 and the 3rd Table in Section 3.5).

**Figure 1 diagnostics-13-00861-f001:**
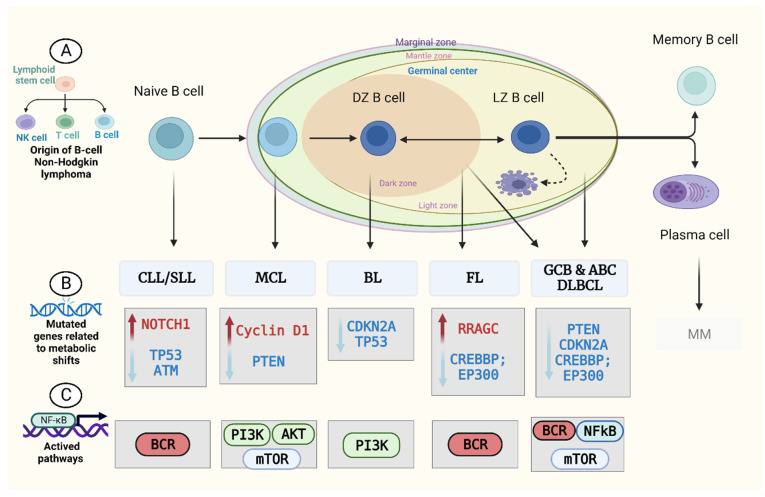
Altered gene expression and mutations associated with key metabolic pathways found in B-NHL subtypes. The figure illustrates: (**A**) The major B-cell non-Hodgkin’s lymphoma subtypes that emerge from different cells that originate within the lymph node; (**B**) mutated genes that influence metabolic reprogramming; and (**C**) critical metabolic pathways observed in B-NHL subtypes. The references used for this figure are CLL/SLL [33,34,35], MCL [36], BL [37], FL [38], and DLBCL [33,39].

### 2.1. Diffuse Large B-Cell Lymphoma (DLBCL)

The most prevalent B-cell non-Hodgkin’s lymphoma is DLBCL. Over 40% of DLBCL patients are refractory and have a worse prognosis for survival [40]. The International Prognostic Index (IPI) is currently used as the primary risk-stratification tool for prognosis in the clinic, and higher IPI scores indicate a worse outcome [41]. However, IPI cannot identify high-risk individuals [42]. Multiple investigations have failed to replicate the predictive power of the molecular heterogeneity of DLBCLs [43], which is widely regarded as a crucial factor influencing the response to therapy [42]. Therefore, additional research is required to identify new prognostic biomarkers to enhance the current DLBCL stratification system and direct the optimisation of therapeutic strategies.

Increased uptake of the glucose analogue 18F-fluoro-2-deoxy-D-glucose (18F-FDG) and up-regulated expression of GLUT1 and HK are indicative of the robust metabolism of DLBCL cells [41,44]. Metabolic heterogeneity, as shown by malignancies’ varied substrate dependency, is common among tumour types and subtypes [4]. DLBCL is metabolically heterogeneous and categorised into oxidative phosphorylation (OxPhos) and BCR groups [20,45]. Few studies to date have identified specific metabolic indicators involved in the diagnosis and prognosis of DLBCL (the 3rd Table in Section 3.5); the reader is directed to previous studies on these topics [46,47,48,49,50,51,52,53,54,55,56,57,58].

### 2.2. Follicular Lymphoma (FL)

An indolent lymphoma originating from germinal centre B cells is called follicular lymphoma (FL) [59]. It is the second most prevalent lymphoid malignancy, and accounts for 20% of non-Hodgkin’s lymphomas and is a disease of adults [60]. Transformation into DLBCL is related to increased glycolytic enzyme expression, which is in line with higher glucose uptake by 18F-FDG PET/CT [61]. Banoei et al. found higher levels of ADP, AMP, GTP, NADHP, glucose, and uridine diphosphate glucose (UDP-glucose) in FL compared with controls; and this was linked to aggressive cases of FL [62]. Regrettably, little is known about the metabolism of FL.

### 2.3. Mantle Cell Lymphoma (MCL)

MCL represents about 5–10% of B-NHLs. MCL is classified as indolent, but the disease progresses quite aggressively [63]. Many studies have pointed to a disruption of the upstream PI3K/AKT pathway as a driver of mTOR in MCL. Supporting this idea is the finding that PTEN, an intrinsic PI3K/AKT inhibitor, is often absent or at low levels in MCL [64]. Evidence from clinical trials shows that mTOR inhibition effectively targets MCL metabolism, and so it is authorised for the relapsed/refractory (r/r) setting [61,65]. Glycolysis, PPP, and lipid biosynthesis are all stimulated by mTOR signalling [66]. Higher quantities of lactic acid, TCA metabolites, and amino acids were found in MCL compared with controls, which may suggest a cancer-specific energy metabolic mechanism to ensure ongoing proliferation within the constrained resources of their microenvironment, as reported by Sekihara et al. [64]. By analysing the metabolic processes of MCL cells and their response to the Bruton’s tyrosine kinase (BTK) inhibitor ibrutinib (IBR), Lee et al. proposed imaging biomarkers (lactate and alanine) to detect response and resistance to IBR in MCL and suggested pathways to overcome IBR resistance, most notably glutaminolysis, the major oxidative ATP-producing pathway in these cells [67].

### 2.4. Burkitt Lymphoma (BL)

Dennis Burkitt discovered the rare and aggressive Burkitt lymphoma (BL) [2]. BL is characterised by chromosomal rearrangements of the c-Myc proto-oncogene, which stimulates the expression of multiple enzymes in serine biosynthesis [46]. Serine is necessary for one-carbon metabolism and nucleotide synthesis [68]. Yang et al. studied BL mice serum metabolomics. Glucose, glutamate, and unsaturated lipids were significantly different in BL and controls. Abnormal metabolism and metabolites of BL were found. These discoveries may help create noninvasive approaches for BL diagnosis and prognosis based on these biomarkers [69].

### 2.5. Chronic Lymphocytic Leukaemia (CLL)

Chronic lymphocytic leukaemia (CLL) is characterised by the heterogeneous malignant proliferation of mature monoclonal B cells in the blood, bone marrow, and lymphoid organs [70]. Alterations in carbohydrate metabolism, lipid metabolism, and OXPHOS are all part of the dynamic metabolic reprogramming of CLL that occurs at different stages of the tumour [71]. Furthermore, TP53, ATM, and MYC, among others, are tumour-suppressor genes that regulate the metabolic reprogramming that occurs during CLL [21]. CLL cells are highly glycolytic, but not as much as DLBCL cells [72]. CLL cells exploit altered lipid metabolism to promote mitochondrial function via activating STAT3 [73]. High FDG uptake in a PET/CT scan is an indication of a glycolytic phenotype in CLL cells, which may predict Richter’s transformation into an aggressive lymphoma, most often DLBCL [74].

**Table 1 diagnostics-13-00861-t001:** Significant metabolites in common B-NHL subtypes.

B–NHL Subtypes	Metabolites	Study Purpose	Potential Clinical Utility	References
B-cell lymphoma	↑ Uracil	Uracil levels in normal and malignant B cells from mice and humans	Early detection	[75]
FL	↑ ADP, ↑ AMP, ↑ GTP, ↑ NADHP, ↑ glucose, and ↑ UDP-glucose	Metabolomics signatures that distinguish FL from controls	Predictive of outcome	[62]
MCL	↓ lactate and ↓ alanine	Examine ibrutinib’s mechanism of action in MCL cells	Therapeutic monitoring	[67]
BL	↓ Glucose, ↑ glutamine, and ↑ choline	Investigated the serum metabolomics of BL mice models	Diagnosis Prognosis	[69]
CLL	↓ Glucose, ↑ glutathione, ↑ lipid, and ↑ glycerolipid	Investigate miR-125b’s role in CLL	Diagnosis Prognosis	[76]

## 3. Metabolomics and B-NHL Biomarker Discovery

Metabolomics uses nuclear magnetic resonance (NMR) or mass spectrometry (MS) to look at global, dynamic, and endogenous metabolites [77,78]. Metabolomics has been used to explore disease pathogenesis and discover novel biomarkers. Thus, metabolomics can be utilised not just to identify new biomarkers but also to develop noninvasive diagnostic and prognostic tools for medical conditions [69,79]. In the study of cancer, introducing certain novel technologies such as metabolomics is found to impart fruitful and reliable information regarding cancer metabolism, particularly for the main mechanism in tumour proliferation [77]. The impact of B-cell lymphoma on the patient’s metabolomics is still not fully known. Little research has been conducted on the treatment response and prognosis of B-cell lymphoma [80]. When looking for alternative methods to improve the rate of detection and compliance in the assessment of B-cell non-Hodgkin’s lymphoma, the study of metabolomics with its comprehensive and unbiased exploration for changes in the metabolic profile has been found to be an effective approach [12]. Thus, developing metabolomics technology and functional metabolic assessment in B-cell non-Hodgkin’s lymphoma remains an interesting subject for enhanced diagnosis and therapy [23].

As discussed below, there are a variety of steps to metabolomics analysis (Figure 2), each with their own set of benefits and drawbacks [81,82].

### 3.1. Metabolomics Study Design

Metabolomics studies can be divided into two classes: targeted and non-targeted. Targeted analysis is used for the identification and quantification of pre-defined metabolites and can be used for quantitative as well as qualitative analysis [83]. Non-targeted analysis consists of analysing all accessible metabolites in a given sample and is the first choice for cancer biomarker discovery studies [84]. Therefore, non-targeted metabolomics research necessitates advanced analytical methodologies, computerised spectral data processing, biological data elucidation, and hypothesis generation [85,86,87]. In the DLBCL studies, non-targets were the most frequently applied, with an average of 61.5% compared with targeted methods (the 3rd Table in Section 3.5).

### 3.2. Sample Collection and Preparation

The collection of the sample, its preparation, and storage are the second step in the metabolomics study plan. The most common samples for conducting clinical metabolomics research are blood and urine [88]. It is important to design the research based on metabolomics to reduce the influence of certain constituents such as age, gender, state of fasting, diet, physical activity, exercise, and the day and time of sample collection. Before starting the actual research, it is important to conduct a pilot study of healthy individuals and report it as part of the research to validate the results’ reproducibility. The samples (particularly plasma, serum, and urine) must be kept in various aliquots soon after collection to avoid the production of compounds from the many freeze–thaw cycles used for different metabolomics studies [89]. The factors used for processing the sample, such as pH buffering and extraction, should also be uniform and follow standard operating procedures (SOPs) [81,82,90,91]. The samples that are non-invasive in nature, such as blood or urine, are the best for regular clinical analysis [85]. Comparing the serum metabolomics of high-risk individuals with those who have been cured by standard chemotherapy can provide useful information about the prognosis of DLBCL as well as the mechanisms involved in failed treatment procedures [92,93]. For best practices in metabolomics, the reader is directed to previous reviews on these topics [94,95,96,97].

### 3.3. Analytical Techniques

The study of metabolomics is regarded as one of the most trustworthy and comprehensive tools for investigating the physiological parameters of an individual, analysing the metabolic pathways, and discovering new biomarkers [98] by employing mass spectrometry (MS), and nuclear magnetic resonance (NMR) spectroscopy technologies [85,92]. 

#### 3.3.1. LC–MS

The MS technique has the ability to isolate the intricate mixture of compounds for their detection and quantification with elevated sensitivity and specificity, and can also demonstrate information regarding molecular structures [99]. MS separation techniques are essential for reducing sample complexity and minimising ionisation suppression effects [100]. A preceding separation stage, such as high-performance liquid chromatography (HPLC), or ultra-performance liquid chromatography (UPLC), and capillary electrophoresis (CE), is frequently required. There are three main components in a mass spectrometer: an ion source, a mass analyser, and a detector. The ion source is used for converting the sample molecules into ions, which are then resolved into an electromagnetic field or time-of-flight tube by the mass analyser, while the detector is employed for measuring the end results. For maximising the coverage of the metabolome, it is advisable to conduct the analysis of biological samples in the *m*/*z* 50–1000 scan range and in both positive and negative ionisation forms [101,102]. Electrospray ionisation (ESI) is used in metabolomics trials due to its “soft ionization” competency and ability to produce unbroken molecular ions [102]. 

Medriano et al. examined the metabolomics of two types of blood cancers, myeloma and non-Hodgkin’s lymphoma, using plasma samples from both cancer patients and healthy individuals to detect all the potential metabolites and pathways that were affected by employing metabolomics based on LC–MS. Their results revealed a significant metabolomic difference between the healthy control individuals and the myeloma and NHL patients, with disturbed metabolic pathways such as choline metabolism and oxidative phosphorylation being associated with the progression and growth of tumours [77].

#### 3.3.2. GC–MS

GC–MS is a technique that combines great separation efficacy with sensitive, selective, and versatile mass evaluation and is suitable for comprehensive analysis. It is a combination of MS and GS that is used for the detection and quantification of a wide range of chemical compounds, such as natural products, blood, and urine. GS–MS is used in many fields of study, such as detecting drugs, amino acid evaluation, doping control, and the detection of natural materials like food products [103]. EI, or electron ionization, is used for combining MS with GC in almost all the metabolomics applications that are based on GC. The EI–MS method works well for chemical compounds that do not change when heated and that are volatile and are separated by chromatography at high temperatures [104].

Bueno Duarte et al. collected urine samples from NHL patients and conducted their metabolic analysis by employing untargeted GC–MS, which was found to be a valuable tool for distinguishing the population under study. Their GC–MS results indicated the presence of as many as 18 metabolites in the urine sample that contributed to differentiating healthy subjects from DLBCL patients with an accuracy of about 99.8% (*p* < 0.001). GC–MS is considered a valuable option for studying metabolomics due to its operational simplicity, low cost, reliable identification of metabolites, robustness, and easy availability [105].

#### 3.3.3. NMR

NMR spectroscopy is a universal metabolite detection method that allows for direct analysis of samples with little sample preparation and simultaneous measurement of numerous types of tiny metabolites [106,107]. However, it has limitations, such as high equipment costs, high maintenance costs, and decreased sensitivity [108,109]. Mass spectrometry is better than NMR in several ways, although NMR has its own advantages (Table 2). The B-NHL study design determines the optimal analysis.

**Table 2 diagnostics-13-00861-t002:** Comparison between LC–MS, GC–MS, and NMR platforms.

Characteristics	LC–MS	GC–MS	NMR
Sensitivity	High	High	Low
Reproducibility	Moderate	Low	High
Quantitative analysis	Not very quantitative	Quantitative	Quantitative
Metabolite identification	More (database available)	Few	Limited
Non-destructive sample	No	No	Yes
Sample preparation	Need derivatisation/chemical modification	Requires sample derivatisation	Requires minimum sample preparation
Tissue samples extraction	Required	Required	Not required
Experimental time	Slow	Slow	Fast
Experiment cost	High	Affordable	Low

### 3.4. Data Acquisition and Processing

When the metabolomics data are produced, it is important to ensure that they are reproducible [89,110]. Quality standardisation and quality control are considered for the optimisation of the reproducibility of results. Data analysis and bioinformatics are used to process the data, which are then subjected to statistical analysis. There are two classical approaches to the statistical analysis of multivariate data: unsupervised learning and supervised learning. A popular unsupervised learning method is principal component analysis (PCA). The second main approach is supervised learning, such as with artificial neural networks (ANN), partial least squares discriminate analysis (PLS-DA), etc., which can be used for excavating the data further to obtain the biomarkers [111,112]. The discovery process of biomarkers can be driven through supervised models that can be linked with clinical results, histopathological scores, and various other omics data. It is important to test the supervised models with precise internal cross-validation processes or external tests to obtain trusted biomarkers and models and to decrease the chances of data overfitting [113].

### 3.5. Metabolites Identification: Biomarker Discovery and Validation

Profiling the metabolites in each biological entity is incomplete without accurate data measurement and precise interpretation. To identify the features of potent spectral biomarkers, attempts are made to recognise the unidentified spectral biomarkers. The peaks can be identified with the help of public metabolomics databases and in-house spectral databases such as the Golm database, LIPID MAPS, human metabolome database (HMDB), METLIN database, etc. Following the identification of metabolomics biomarkers, additional experiments are required to validate or test the biomarkers [82,112,114].

DLBCL is the most common non-Hodgkin’s lymphoma, and therefore 13 publications of relevance to our research interest are included in this review (search query in PubMed: “Metabolomics” and “DLBCL”), and summarised in Table 3.

## 4. Applications of Metabolomics in B-NHL

In the clinical setting, metabolomics is finding an expanding number of applications, including disease diagnosis and understanding, the discovery of novel drug targets, the customisation of medication treatment, and the monitoring of therapeutic results [115]. In this last section, we discuss the clinical applications of metabolomics and offer examples to clarify how metabolism will open a new era in lymphoma research and how this will positively influence diagnosis and treatment.

### 4.1. Discovering Targeted Therapies Based on Metabolomics

Metabolism in B-NHL plays a crucial role in established therapeutic approaches (Table 4). Antimetabolites were the name given to the chemical compounds that were first used to treat cancer. The reason for choosing this name was that these compounds were found to resemble endogenous metabolites in their chemical structure and disrupt the process of normal metabolism. In comparison to other omics, metabolomics is best for evaluating the potential of these cancer treatment regimens. This study was carried out to discover whether the therapies could cause alterations in the metabolic pathways and detect the pharmacokinetics of drugs simultaneously or not. In the coming time, it will become crucial to combine the study of pharmacometabolomics with other biological systems knowledge, such as mRNA, genetics, miRNA, and imaging. This will help in determining the correlation of the metabolomics response with the cancer stage, undesirable incidents, and the growth or recession of the tumour. The study of pharmacometabolomics is capable of monitoring a patient’s metabolic response to a drug; thus, it is very interesting to use metabolomics in detecting cancerous growth, prognosis, and therapy management [82]. Moreover, the therapies based on metabolomics can not only enhance the responses of immune cells to extremely immunogenic tumours but can also elevate the immunogenicity of cancer cells, thereby increasing the ability of immunotherapy to cure a vast variety of carcinomas. For further information, the reader is directed to previous reviews on these topics [23,116,117,118,119,120].

### 4.2. Determining B-NHL Diagnostic and Prognostic Biomarkers

A recent metabolomics study suggested a methodology for discovering novel biomarkers that can be used for the diagnosis and characterisation of various lymphoma subtypes. The GC–MS method was used for the investigation and evaluation of plasma samples taken from individuals with different subtypes of lymphomas. The results showed a significant prevalence of elaidic acid and hypoxanthine (HX) in patients suffering from Hodgkin’s lymphoma, MM, CLL, and DLBCL compared with healthy control individuals in all the study groups [50]. Yoo et al. analysed the urine samples taken from lymphoma patients and translated the data into ions of low mass, i.e., less than 1000 *m*/*z*. They chose three peaks of high intensity and low mass ions for the analysis, of which the peak in the range of 137.08 *m*/*z* ion was detected as HX. The levels of HX and xanthine inside the cells are found to be inversely proportional to the energy modifications of adenylate and thus to the ATP of the cells. Additional research is required as abnormal metabolic processes are detected as initial lymphoma biomarkers [127]. For further information, the reader is directed to previous reviews on these topics [53,80,118,128].

### 4.3. Determining the Lymphomagenesis Risk Factors

Genetic mutations accumulate sequentially during tumour development, eventually resulting in malignant tumours. However, it has also been shown that metabolic processes and inflammatory factors indirectly contribute to the development of the tumour [11]. In their study, Pettersena et al. proved that the cell line of B-cell lymphoma surrounds numerous amplified genomic uracil concentrations in comparison with non-lymphoma cell lines or normal lymphocytes. They utilised a method based on liquid chromatography combined with mass spectrometry (LC/MS) for quantifying the genomic sequence of 2-deoxyuridine and proving their study. In harmony with uracil generated by activation-induced cytidine deaminase (AID), they discovered a distinctive mutational signature of an AID hotspot in the lymphoma area where there was clustered mutation. They also presented an important revelation about the expression of SMUG1 and uracil-DNA glycosylases UNG along with the excision capacity of uracil by stating its negative correlation with the concentration of genomic uracil, which somewhat decreased the AID effect [129]. Another study was also conducted on the metabolomic pattern of Burkitt lymphoma that was induced by MYC glucose deprivation, as well as hypoxic and aerobic conditions. They used a [U-13C, 15N]-glutamine tracer to detect glutamine import and metabolism via the TCA cycle under hypoxia conditions and discovered that glutamine is significantly precipitated to citrate carbons. The deficiency of glucose leads to the significant augmentation of citrate, fumarate, and glutamine-derived malate. Their arrangements showed a different pathway for the generation of energy called glutaminolysis, which is associated with the glucose-independent TCA cycle. Under the conditions of hypoxia and scarcity of glucose, the critical role of glutamine in the proliferation of cells makes them susceptible to BPTES (glutaminase inhibitors), which in turn can be used for treating tumours [130].

## 5. Conclusions

As the use of metabolomics is continuously increasing in clinical trials, it may soon become one of the most successful tools for detecting and healing cancerous growths. The changes related to metabolic pathways may occur in a broad range of B-cell non-Hodgkin’s lymphomas. Researching and knowing about them can help in identifying new remedial targets and discovering novel metabolic biomarkers. In the near future, the study of metabolomics will become crucial for outcome prediction and the revelation of new treatment regimens. There is a need for conducting metabolomics research on B-cell lymphomas in large cohort trials to discover new biomarkers, which will thus prove to be an influential step in the path of clinical integration of biomarkers that are discovered by metabolomics. 

## Figures and Tables

**Figure 2 diagnostics-13-00861-f002:**
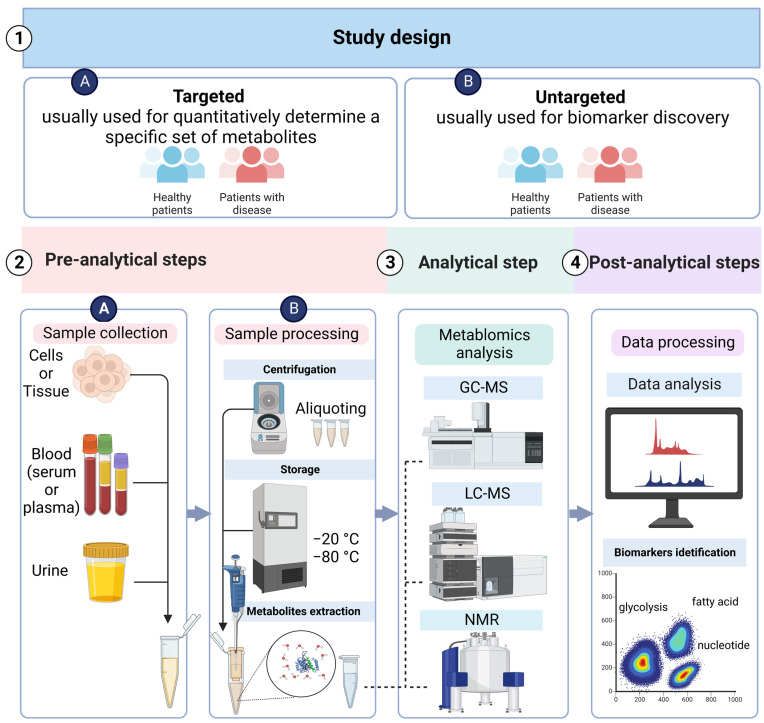
B-NHL Metabolomics Workflow Steps: (**1**) study design; (**2**) pre-analytical process, including sample collection and processing; (**3**) analytical process, which is platform choice (either LC–MS, GC–MS, or NMR); and (**4**) post-analytical process, including data processing, results interpretation, and biomarker identification.

**Table 3 diagnostics-13-00861-t003:** Metabolic markers of diagnostic and prognostic significance in DLBCL.

Metabolic Markers	Study Design	Sample Type	Analytical Platform	Statistics	References
Alanine, aspartate, glutamate, cysteine, & methionine	Untargeted	Cell lines	UHPLC/MS	*t*-test & partial least square discriminant analysis (PLS-DA)	[47]
Asparagine & serine	Targeted	Cell lines	NMR	Two-sided Fisher’s exact test & principal component analysis (PCA)	[46]
lysine & arginine	Untargeted	Serum	NMR	Supervised multivariate analysis	[48]
Valine, hexadecenoic acid & pyroglutamic acid	Untargeted	Serum	GC/MS	PCA & PLS-DA	[49]
2-aminoadipic acid, 2-aminoheptanedioic acid, erythritol & threitol	Untargeted	Plasma	GC/MS	*t*-test, multivariate analyses & PLS-DA	[50]
Ornithine	Untargeted	Cell lines	GC/MS	*t*-test, one-way ANOVA) & orthogonal partial least-squared discrimination analysis (OPLS-DA)	[51]
Pyruvic acid	Targeted	Cell lines & FFPE	NMR & GC/MS	The Shapiro–Wilk test, two-sided Welch test, the nonparametric Mann–Whitney U test & PCA	[52]
Malate	Untargeted	Plasma	GC/MS	two-tailed Student’s *t*-test, one-way ANOVA, PCA, a supervised PLS-DA & OPLS-DA	[53]
2-arachidonoylglycerol (2-AG)	Untargeted	Serum & cell lines	HPLC/MS	Two-tailed *t*-test, and XCMS/R	[55]
Lactate	Targeted	Cell lines	GC/MS	Two-tailed *t*-test, Kaplan–Meier curves & log-rank test	[54]
Glycine	Targeted	Cell lines	HPLC/MS	*t*-test	[56]
Choline	Targeted	Serum	UPLC/MS	Two-tailed *t*-test	[57]
Choline	Untargeted	Plasma	UHPLC/MS & GC/MS	*t*-tests & supervised multivariate analysis	[58]

**Table 4 diagnostics-13-00861-t004:** Therapeutic drugs for B-NHL metabolism.

Agents	Target	Status	Tumour Effect	References
Ritonavir + metformin	GLUT4+ETC inhibition	Approved for non-malignant indication	CLL cell death	[121]
Idelalisib	PI3Kδ inhibition	Approved	CLL and FL cell death	[122,123]
Ibrutinib	BTK inhibition	Approved	CLL and MCL proliferation inhibition	[124,125]
AZD3965	MCT1/MCT2 inhibition	Phase I trial currently running	DLBCL and BL proliferation inhibition	[126]

## Data Availability

Not applicable.

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
