# Peer review of "Metabolomics: A New Era in the Diagnosis or Prognosis of B-Cell Non-Hodgkin’s Lymphoma"

_diagnostics, 2023, doi:10.3390/diagnostics13050861_

Round 1

Reviewer 1 Report

In the present review Alfaifi and colleagues the potential impact of metabolomics in B cell lymphomas. 

Probably the authors may reduce the “technical” parts that compose the section “3. Metabolomics and B-NHL Biomarker Discovery”

Few additional comments: 

·      Page 3, line 7: amplification could also affect the MYC gene in lymphoma; please quote them. 

·      Figure 1: In panel B the term “mutated genes” is not correct since most of MYC abnormalities are not gene mutations but chromosomal translocation/amplification. In addition, in CLL MYC is rarely mutated/rearranged. Please improve this figure.  

·      Page 5, line 39-42: What do the authors mean for “pyruvate and glutamate levels may be useful indicators of CLL patients”?

·      Table 1: The authors need to clearly explain in which way metabolites may be useful in helping diagnosis since today lymphoma diagnosis is based on histology. 

·      Page 6, line3: thus, should be read Thus and Metabolomics should be read metabolomics. 

·      Page 7, line1: Targeted should be read targeted.

Author Response

Response to Reviewer 1 Comments

Round 1

Point 1: Probably the authors may reduce the “technical” parts that compose the section “3. Metabolomics and B-NHL Biomarker Discovery”

Reply 1: we completely agree with you the technical part is too long and it describes many superficial details, so we have cut back (trim) the technical part.

Point 2: Page 3, line 7: amplification could also affect the MYC gene in lymphoma; please quote them.

Reply 2: we have done.

Point 3: Figure 1: In panel B the term “mutated genes” is not correct since most of MYC abnormalities are not gene mutations but chromosomal translocation/amplification. In addition, in CLL MYC is rarely mutated/rearranged. Please improve this figure.

Reply 3: we have improved the figure 1, and cited the references.

Point 4: Page 5, line 39-42: What do the authors mean for “pyruvate and glutamate levels may be useful indicators of CLL patients”?

Reply 4: we have removed it, because that does not add any new information.

Point 5: Table 1: The authors need to clearly explain in which way metabolites may be useful in helping diagnosis since today lymphoma diagnosis is based on histology.

Reply 5: We have improved Table 1, and also added this paragraph:

There is an urgent need for biomarkers based on non-invasive sampling procedures (e.g., blood, urine, etc.) that can help in the diagnosis of lymphoma, such as metabolite profiling. The perfect test should be easy, reliable, and accurate. "What simple, non-invasive, painless, and convenient tests can be used to detect cancer early?" ranked as the most important research priority for the early detection of cancer in the UK-focused research gap survey performed by the James Lind Alliance alliance, which includes patients and doctors [31]. by Accordingly, serum biomarkers of lymphoma activity have been studied extensively over the last decade [32], and we conclude that they are clinically relevant for the diagnosis, prognosis, and therapeutic monitoring of lymphomas. In this review, we shed light on the most major metabolic dysregulation described in B-cell non-Hodgkin lymphoma research (tables 1 and 3). “ page 5.

Point 6: Page 6, line3: thus, should be read Thus and Metabolomics should be read metabolomics.

               Page 7, line1: Targeted should be read targeted.

Reply 6: we have corrected.

Reviewer 2 Report

Thanks for submitting the research topic “METABOLOMICS: A NEW ERA IN THE DIAGNOSIS OR PROGNOSIS OF B CELL NON-HODGKIN’S LYMPHOMA”. There is no deny that this manuscript present in detail the development of metabolomics in the diagnosis and prognosis of B cell non-Hodgkin’s lymphoma. However, this review resembles textbook for describing many superficial details, especially in the part of “Analytical Techniques”, while lacking in sufficient summary of the researches in cutting-edge areas, for examples, the innovation and limitation of a particular research or the future direction. Hope you authors can trim this manuscript.

Author Response

Response to Reviewer 2 Comments

Round 1

Point 1: this review resembles textbook for describing many superficial details, especially in the part of “Analytical Techniques”, while lacking in sufficient summary of the researches in cutting-edge areas, for examples, the innovation and limitation of a particular research or the future direction. Hope you authors can trim this manuscript.

Reply 1: we completely agree with you the technical part is too long and it describes many superficial details, so we have cut back (trim) the technical part.

On the other hand, this review aims to shed light on the metabolomics technique, so the review covers biological and technical aspects to be a reference for those interested in this field. Previsuly we have published paper about “Metabolic Biomarkers in B-Cell Lymphomas for Early Diagnosis and Prediction, as Well as Their Influence on Prognosis and Treatment”, so that paper describe the innovation and limitation in this field, we referred for that in last section 4.2 referance (78), page 17.
